# An Enveloping, Centering, and Grabbing Mechanism for Harvesting Hydroponic Leafy Vegetables Cultivated in Pipeline

Yuanjie Liu, Hongmei Xia *, Junjie Feng, Linhuan Jiang, Liuquan Li, Zhao Dong, Kaidong Zhao and Jiamou Zhang

College of Engineering, South China Agricultural University, 483 Wushan Road, Guangzhou 510642, China
* Correspondence: xhm_scau@scau.edu.cn

**Abstract:** Manually harvesting hydroponic leafy vegetables from a cultivation pipeline is labor-intensive and expensive. Rapidly grabbing hydroponic leafy vegetables grown in different positions and orientations in the planting hole is the primary issue for efficient mechanical harvesting. Thus, a novel grabbing mechanism with double-pivot rotation cross fingers is proposed. The fingers' inner surfaces could envelop the grabbable area of the leafy vegetable in the grasping process and position each leafy vegetable stalk to the center of the planting hole before taking it out. A grabbing mechanism for harvesting hydroponic Chinese kale was designed and optimized with less than 1 mm of grasping error, enough enveloping range, and no collision with the extended leaves. Laboratory experiments were conducted to investigate centering and grabbing at different initial positions and inclination angles of the hydroponic Chinese kale and varied finger deflection speeds. It was indicated that for the grasping inclination angle and grabbing success rate, the initial inclination angle was a significant factor, as was the position, whereas the finger deflection speed was insignificant. As the initial inclination angle of matured hydroponic Chinese kale in different initial positions is mostly larger than 60°, the best results were achieved with a finger deflection speed in the range of 40° s$^{-1}$ to 60° s$^{-1}$ and grasping inclination angles of 85° to 95°, with a grabbing success rate of more than 95%. This showed the promising applicability of the studied grabbing mechanism for harvesting hydroponic Chinese kale or other varieties of leafy hydroponic vegetables with similar growth characteristics.

**Keywords:** hydroponic leafy vegetable; harvesting; hydroponic Chinese kale; grabbing mechanism

## 1. Introduction

A shallow liquid flow pipeline is a clean and pollution-free, water- and fertilizer-conserving, high-quality and high-yield planting method widely used in the hydroponic production of leafy vegetables in domestic and international greenhouse facilities [1–4]. Efficient and automatic non-harvesting equipment for seeding, seedling, and transplanting operations have already been applied in hydroponic leafy vegetables production [5,6]. Only auxiliary conveyors were equipped for grasping leafy vegetables and removing them from the cultivation pipelines by the manual method [7,8]. Manual harvesting is labor-intensive, costly, and inefficient, requiring a large number of workers for timely harvesting [9]. With the development of Chinese industrialization, more and more rural laborers have migrated into urban areas. Thus, the shortage of agricultural workers and high labor costs have created serious issues for planting farmers. Therefore, developing equipment for harvesting hydroponic leafy vegetables is crucial for improving the productivity and economic benefits of hydroponic planting enterprises and promoting the development of a sustainable leafy vegetable hydroponic industry [10].

Previous studies on hydroponic leafy vegetable harvesters were designed for planting plate cultivation mode. In those studies, roots were cut by a relative motion between the leafy vegetables and the cutter; then, the roots and edible stalks were separately delivered by conveyors and gathered by the collecting mechanisms. For example, with the harvester

developed by Netherlands Viscon Company, mature hydroponic leafy vegetables in the planting plate were transported to the root cutting device, roots of the moving hydroponic leafy vegetables were cut by a fixed cutter, conveyed to the collecting box, and edible leaves left in the planting plate were taken out and packed by workers. Gao et al. studied an automatic harvesting device for Chinese cabbage aerosol cultivation which could first hold the Chinese cabbage, then cut its root from the top of the planting plate, and, finally, collect the root and leaves separately [11]. Mo et al. investigated an automated harvesting device for hydroponic lettuce that could cut the roots from the bottom of the planting plate and then gather the root-cut lettuce by a grabbing mechanism [12]. Ma et al. designed a low-damage harvester for hydroponic lettuce that utilized a planting plate with grooves around the planting holes to place the clamping rods. Lettuce could be held by the clamping rods first, and then the roots could be cut from the top of the planting plate [13]. Li et al. put forward a Shanghai cabbage harvesting mechanism that used a band saw to cut roots first and then the collection device to gather the root-cut Shanghai cabbages [14].

Few studies on hydroponic leafy vegetables cultivated in pipelines have been reported. Xu et al. proposed a simple and efficient greenhouse harvesting machine that could cut the bottom stalks of leafy vegetables with a rotating cutter from the top of the cultivation pipeline and then send the root-cut vegetables to a conveyor belt by a pushing plate [15]. However, it is difficult to accurately position the cutter on the stalk, as leafy vegetable roots unconstrained by planting cups grow in different positions and orientations in the planting hole, and the cut roots remaining in the cultivation pipeline require subsequent clearance. Cho et al. designed a harvesting robot with three degrees of freedom for lettuce grown in the planting hole with a position control of planting cups, which first obtained the lettuce height and position information with a machine vision system and then conducted a grasping and cutting operation by an end effector. Its cutting position accuracy was high while the lettuce roots were still in the pipeline [16].

In the mass production of leafy vegetables cultivated in a pipeline, seedlings are usually transplanted into the planting hole of the cultivation pipeline without using planting cups for position control to reduce the material cost and cleaning of the planting cups. As the root diameters of mature hydroponic leafy vegetables are much larger than the stalk diameters, the diameter of the planting holes on the cultivation pipeline is about 2 to 3 times the root diameter of mature leafy vegetables to avoid interference during transplantation and harvest. For hydroponic leafy vegetables cultivated without planting cups, their bottom stalks grow close to the edge of the planting holes with an inclination angle relative to the center of the planting hole, and their roots have a certain bending degree relative to the stalks. When directly grasping the bottom stalk of a hydroponic leafy vegetable and picking it up, the root unconstrained by the planting cup is most likely constrained by the planting hole. Thus, for successfully harvesting hydroponic leafy vegetables grown without the constraint of a planting cup, accurately positioning their bottom stalks is critical.

In previous precision grabbing research, visual methods were applied first to gain information on the position of fruits or vegetables, and then the grabbing mechanisms were adjusted to harvest them in different positions and inclination angles [17–22]. The grabbing process based on vision technology shown in Figure 1 illustrates (a) recognizing and locating the grabbable area, (b) adjusting the position and orientation of the gripper, (c) driving the gripper to the grabbable area of the leafy vegetable, (d) closing the gripper to grasp the grabbable area of the leafy vegetable, (e) adjusting the gripper to align the center of the leafy vegetable stalk with the center of the planting hole, and (f) drawing the gripper from the center of the planting hole in the upward direction to remove the root of the leafy vegetable. The vision-technology-based grabbing method would need to adjust the position and orientation of the gripper repeatedly and would result in a complicated and low-efficiency harvest.

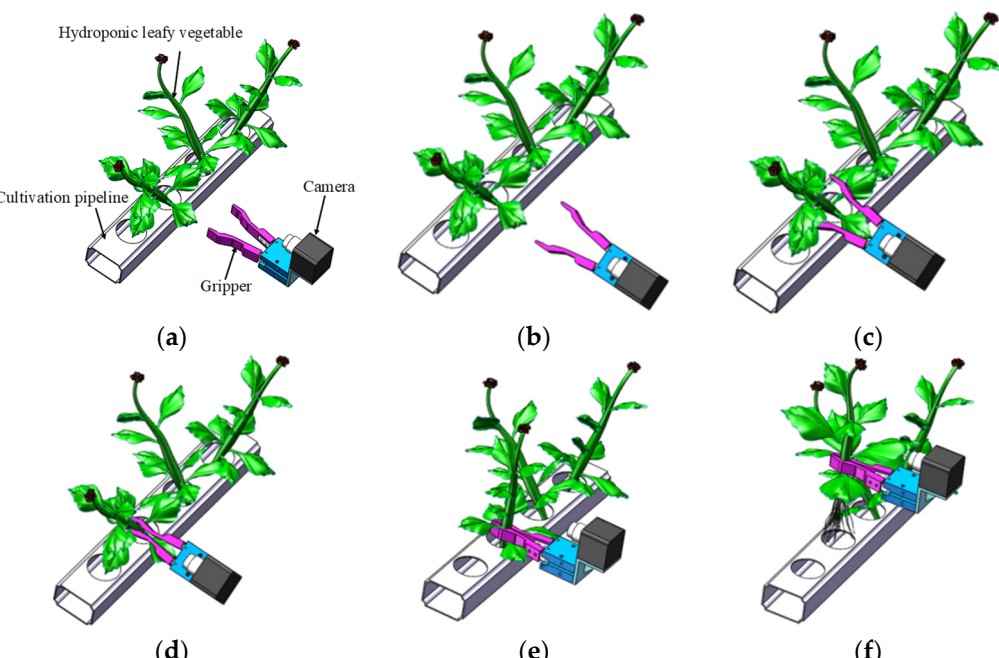

**Figure 1.** Vision technology process of grabbing a leafy hydroponic vegetable: (**a**) recognize and locate the grabbable area, (**b**) adjust the position and orientation of the gripper, (**c**) drive the gripper to the grabbable area, (**d**) close the gripper to grasp the grabbable area, (**e**) adjust the gripper to align the center of the leafy vegetable stalk with the center of the planting hole, and (**f**) draw the gripper along the center direction of the planting hole to move the root of the leafy vegetable out of the planting hole.

As the shape, size, position, and orientation of each hydroponic leafy vegetable in the planting hole are different, quickly grasping the vegetable stalk at the center of the grabbable area, aligning the center of the vegetable stalk with the planting hole center, and then taking the vegetable out of the planting hole are the key steps in achieving efficient and accurate harvesting of hydroponic leafy vegetables cultivated in a pipeline without planting cups. Hydroponic Chinese kale is the main hydroponic leafy vegetable variety in South China [23,24]. To achieve its efficient centering and grabbing at any position and orientation of the planting hole, the objective of the study was twofold: (1) to design a grabbing mechanism that could position each kale stalk at the center of the planting hole and draw the root out without the interference of the planting hole, and (2) to evaluate the grasping inclination angle, grabbing success rate and the finger deflection speed of the grabbing mechanism with different inclination angles and the initial position of the kale in the planting hole. The studied solution for the centering and grasping mechanism could provide the foundation for the research and development of hydroponic leafy vegetable harvesting equipment.

## 2. Materials and Methods

### 2.1. Hydroponic Chinese Kale Cultivated in Pipelines

Lvbao hydroponic Chinese kale planted in Lvyin Agricultural Technology Development Co., Ltd., Guangzhou, China, was selected as the harvesting material. As shown in Figure 2a, the width and height of the cultivation pipeline are 80 mm and 50 mm, respectively. The diameter of the planting hole on the cultivation pipeline is 50 mm, and the center distance between adjacent planting holes is 100 mm. On the basis of the harvesting grade standard formulated by the Ministry of Agriculture of the People's Republic of China, Lvbao hydroponic Chinese kale should be harvested at the fourth leaf above the root [25]. Therefore, the grabbable kale area is between the first and the fourth leaf above its root, as shown in Figure 2b.

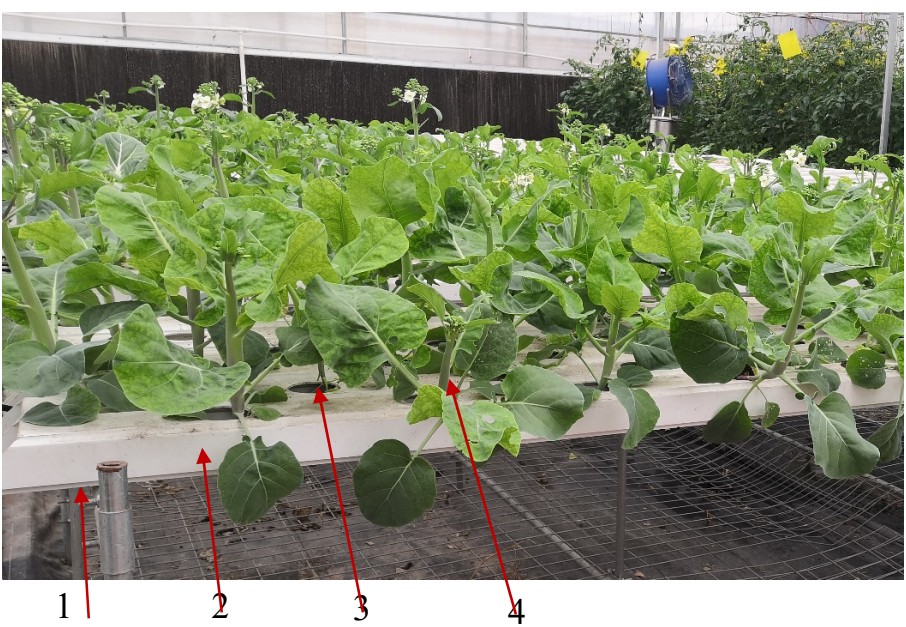

(**a**) 1. Planting bed, 2. Cultivation pipeline, 3. Planting hole, 4. Hydroponic Chinese kale.

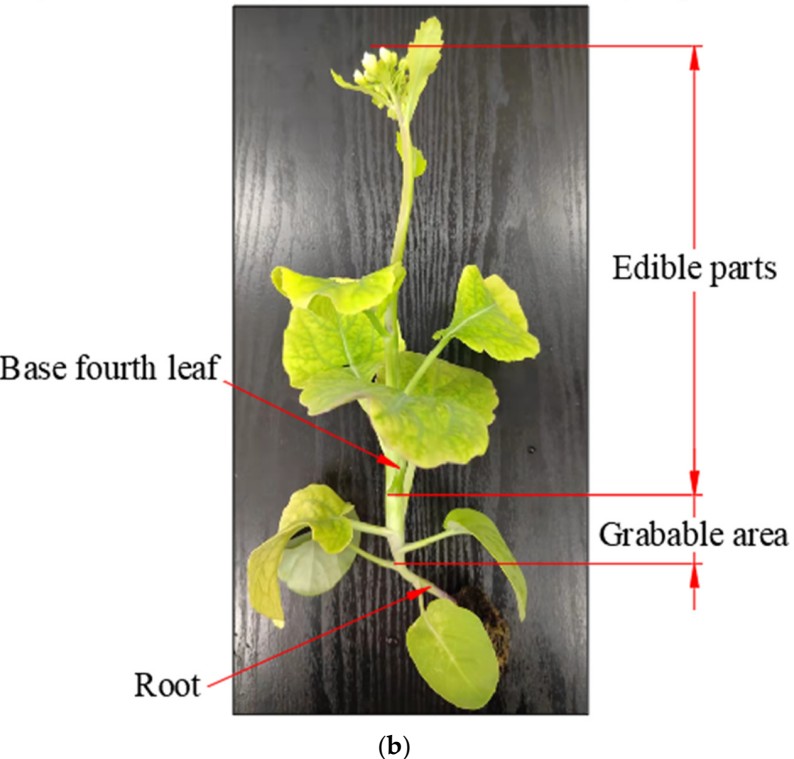

(**b**)

**Figure 2.** Growing conditions of hydroponic Chinese kale: (**a**) the hydroponic Chinese kale planted in the greenhouse cultivation pipeline; (**b**) the main structural features of hydroponic Chinese kale.

The grasping parameters of fifty hydroponic Chinese kale grown for 45 days were measured. Digital electronic vernier calipers (range: 200 mm; accuracy: 0.01 mm) were used to measure the median diameter of the grabbable area. A digital height ruler (range: 500 mm; accuracy: 0.05 mm) was selected to measure the area height. A high-precision electronic scale (range: 1000 g; accuracy: 0.01 g) was utilized to measure the total mass of the hydroponic Chinese kale. The statistical results are shown in Table 1.

**Table 1.** Measurement results of grasping related parameters of Lvbao Chinese kale.

| Parameter | Maximum | Minimum | Mean Value | Standard Deviation | Coefficient of Variation |
|---|---|---|---|---|---|
| The median diameter of the grabbable area (mm) | 16.60 | 7.20 | 11.06 | 2.25 | 0.20 |
| Height of grabbable area (mm) | 42.59 | 4.01 | 25.03 | 8.40 | 0.34 |
| Total mass (g) | 93.80 | 39.30 | 65.70 | 12.22 | 0.19 |

*2.2. Grabbing Mechanism*

2.2.1. Conceptual Design

To analyze the position-adjusting motion function for the extraction of hydroponic Chinese kale, the planting hole coordinate system $O_gX_gY_gZ_g$ and the kale coordinate system $O_yX_yY_yZ_y$ were defined, as shown in Figure 3. Taking the center of the triaxial orthogonal coordinate system $O_gX_gY_gZ_g$ of the planting hole on the upper surface of the cultivation pipeline, the coordinate origin $O_g$ was established, with the $Y_g$ axis along the length direction of the cultivation pipeline, the $Z_g$ axis perpendicular to the upper surface of the cultivation pipeline, and the $X_g$ axis perpendicular to the $Y_g$ and $Z_g$ axes. The triaxial orthogonal coordinate system $O_yX_yY_yZ_y$ was established using the mass center of the hydroponic Chinese kale as the coordinate origin $O_y$, with the $Z_y$ axis following the stalk center direction of the hydroponic Chinese kale and $X_y$ and $Y_y$ axes passing through the origin $O_y$, and perpendicular to the $Z_y$ axis.

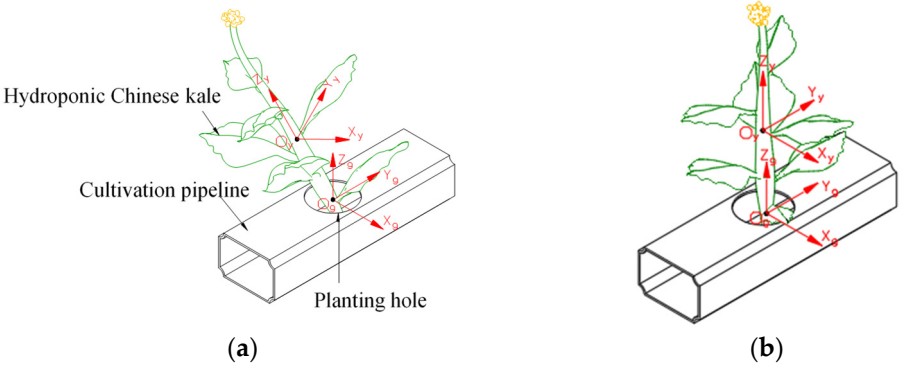

(**a**)          (**b**)

**Figure 3.** Hydroponic Chinese kale in the planting hole: (**a**) the natural growing kale posture in the planting hole; (**b**) the ideal kale posture for removal from the planting hole.

A random growth posture of hydroponic Chinese kale in the planting hole is given in Figure 3a. For successfully grabbing the kale out of the planting hole, its posture should be adjusted to the ideal state, as shown in Figure 3b. Therefore, the grabbing mechanism should have motion control functions for moving the hydroponic Chinese kale to translate along the $X_g$, $Y_g$, and $Z_g$ axes and rotate around the $X_g$ and $Y_g$ axes.

As the shape of the grabbable area of the hydroponic Chinese kale is an approximately symmetrical rotational body, the grabbing mechanism configuration of two V-shaped fingers was designed, which could secure contact with the grabbable area of the hydroponic Chinese kale at any edge position of the planting hole. The clamping process of the two V-shaped fingers on the leafy vegetable is shown in Figure 4. The inner sides of the two V-shaped fingers could apply normal contact on the stalk of the leafy vegetable in the $X_gO_gY_g$ plane. The root of the leafy vegetable could be constrained along the $Z_g$ axis by the inner bottom of the cultivation pipeline. Therefore, the two V-shaped fingers could gradually enclose the grabbable area of the hydroponic Chinese kale and move it to the center of the planting hole.

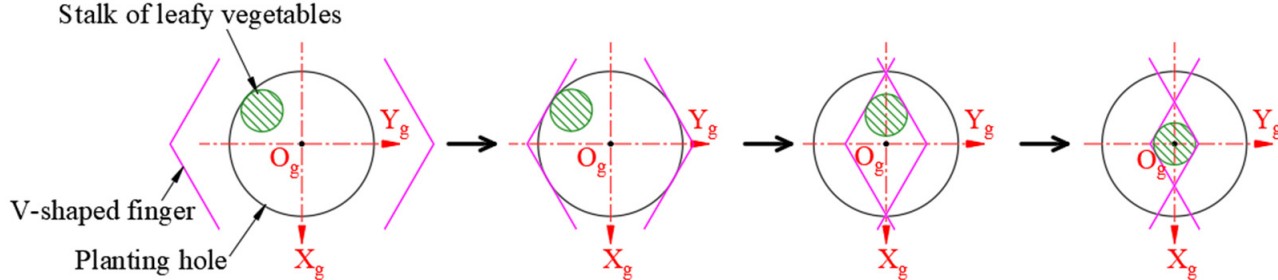

**Figure 4.** Clamping process of the two V-shaped fingers on the stalk of leafy vegetables.

In order to adjust the center of inclined hydroponic Chinese kale to the center of the planting hole, suitable torque should be applied to the kale grabbable area, and interference of the two fingers needs to be avoided during the clamping process. For this reason, the two V-shaped cross fingers were designed as a three-layer structure along the $Z_g$ axis, as shown in Figure 5. One layer is a solid structure single finger. The other two layers are formed by the second finger's hollow structure through which the single finger passes in the clamping process. In Figure 5, $P_1$, $P_2$, and $P_3$ represent the single-finger contact surface with the grabbable area of the hydroponic Chinese kale and the upper and lower layers of the double-finger contact surface with the grabbable area of the hydroponic Chinese kale, respectively.

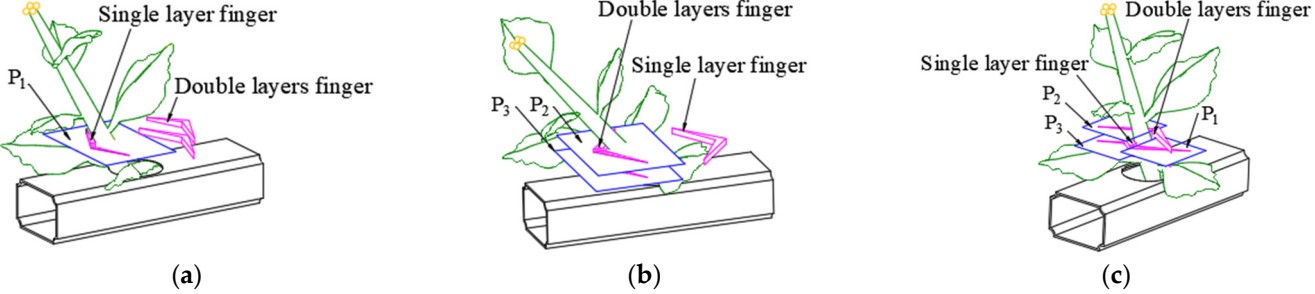

**Figure 5.** Three-layer contact surfaces of two V-shaped cross fingers: (**a**) illustrates the middle sing-layer finger contact surface, (**b**) illustrates the double-layer finger contact surfaces, and (**c**) is the cross-clamping illustration of the two fingers.

The posture adjustment principle of two V-shaped cross fingers is shown in Figure 6, where $F_{P_1}$, $F_{P_2}$, and $F_{P_3}$ represent the respective contact forces of the single-layer finger and both upper and lower layers of the double-layer finger on the grabbable area of the hydroponic Chinese kale. In Figure 6a, the single-layer finger and the lower layer of the double-layer finger are in contact with the grabbable area of the hydroponic Chinese kale at $P_1$ and $P_3$, and $F_{P_1}$ and $F_{P_3}$ form a torque in the grabbable area to rotate stalk of the kale toward the centerline of the planting hole. In Figure 6b, the single-layer finger and the upper layer of the double-layer finger are in contact with the grabbable area of the hydroponic Chinese kale at $P_1$ and $P_2$, and $F_{P_1}$ and $F_{P_2}$ also form a torque on the grabbable area to rotate stalk of the kale toward the centerline of the planting hole. Through the contact of the different layers, the V-shaped cross fingers could exert rotational constraints around the $X_g$ and $Y_g$ axes on the grabbable area of the kale. The two opened V-shaped fingers would first reach outside the planting hole to keep the grasping center consistent with the center of the planting hole and then grasp gradually to enclose the hydroponic Chinese kale in any position and orientation to the center of the planting hole.

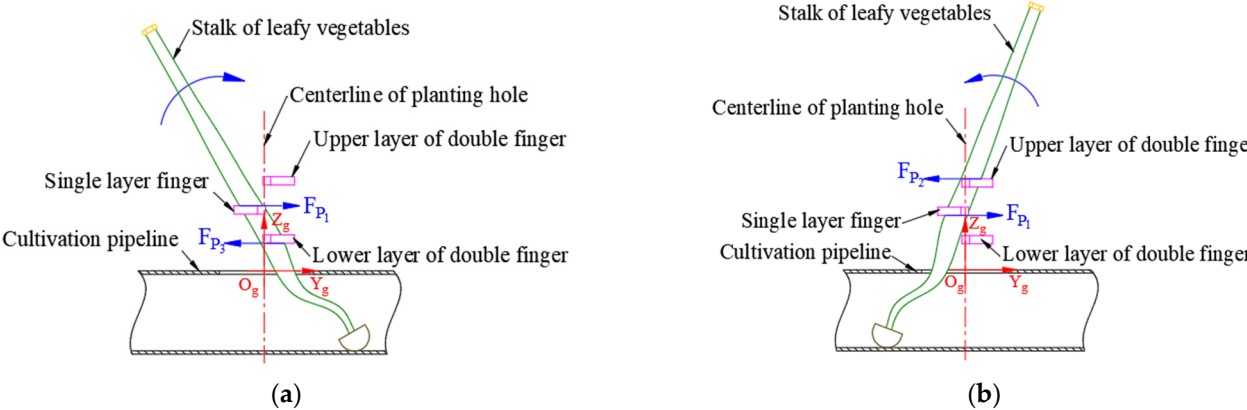

**Figure 6.** Envelope centering and grabbing principle of the two V-shaped cross fingers: (**a**) the centering and grabbing principle of the single-layer finger and the lower layer of the double-layer finger; (**b**) the centering and grabbing principle of the single-layer finger and the upper layer of the double-layer finger.

There are three types of clamping motion of the two V-shaped fingers: parallel translation, single-pivot rotation, and double-pivot rotation, as shown in Figure 7. The theoretical grasping error of the parallel translation pattern is zero. The two parallel open V-shaped fingers would occupy the largest space along the length of the cultivation pipeline and would easily collide with the leaves of the nearby Chinese kale during the reaching process. The margin of error of the single-pivot rotation pattern is larger than that of the double-pivot rotation pattern. For these reasons, the two V-shaped fingers utilized the double-pivot rotation pattern in this study. Because of the diameter differences in the grabbable areas of different hydroponic Chinese kale, the grasping accuracy was achieved by optimizing the main parameters of the two V-shaped fingers.

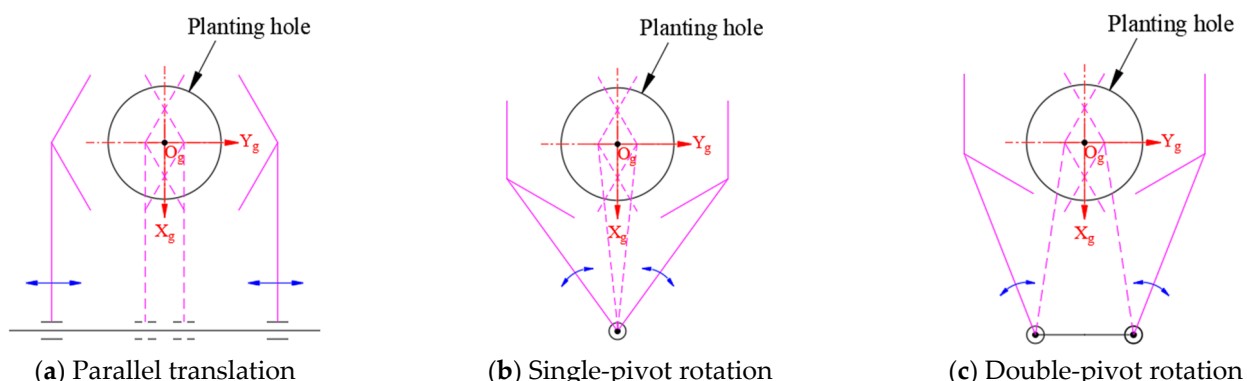

**Figure 7.** Clamping motion pattern of the two V-shaped fingers: (**a**) the parallel translation mode, (**b**) the single-pivot rotation mode, and (**c**) the double-pivot rotation mode.

### 2.2.2. Main Parameters of the Relationships of the Grabbing Mechanism

For its motion symmetry, the two V-shaped fingers with a double-pivot rotation pattern are simplified as a V-shaped finger. When the two open V-shaped fingers reach the grasping position, the main structural parameters of the fingers, the Chinese kale, and the cultivation pipeline are illustrated in Figure 8. The radius of the planting hole is $r_g$, the distance between two adjacent planting holes is $W_g$, the width of the cultivation pipeline is $L_g$, the finger length is $l_{AB}$, the V-slot angle is $2\theta$, the length of the two V-slot sides BC and BD are equal and are represented as $L_v$, the angle between AB and angle bisector of the V-slot is defined as the finger deflection angle $\beta$, the finger rotation angle in the clamping process is $\gamma$, the distance of the double rotary pivot point is $2a$, the distance between

planting hole center and the rotary pivot point is $L_d$, and the radius of the grabbable area is $r_c$.

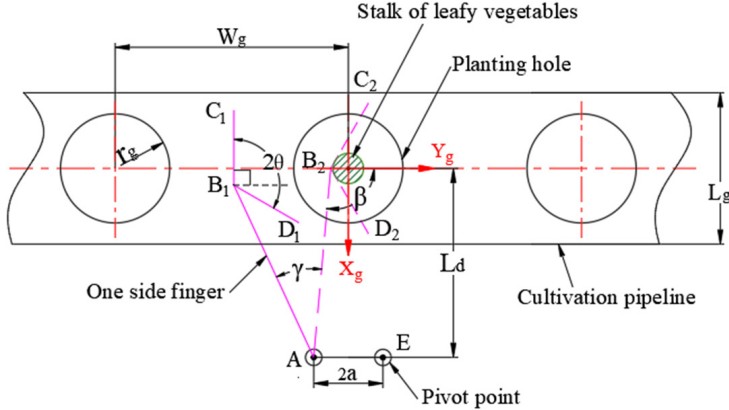

**Figure 8.** Parameters of the double-pivot rotation finger.

To avoid collision with the extended leaves of the Chinese kale, the open BC edge at the front of the finger was along the $X_g$ axis during the reaching process, thus:

$$\gamma = 90° - \theta \tag{1}$$

In Figure 8, we can obtain the following relationships of $L_d$ and $r_c$:

$$L_d = \sqrt{l_{AB}{}^2 + \left(\frac{r_c}{sin\theta}\right)^2 - 2l_{AB}\frac{r_c}{sin\theta}cos\beta - a^2} \tag{2}$$

$$\frac{L_d^2}{l_{AB}{}^2 sin^2\beta - a^2} - \frac{(r_c - l_{AB}sin\theta cos\beta)^2}{sin^2\theta\left(l_{AB}{}^2 sin^2\beta - a^2\right)} = 1 \tag{3}$$

when $r_c = l_{AB}sin\theta cos\beta$, the minimum value point of $L_d$ is:

$$L_{dmin} = l_{AB}sin^2\beta - a^2 \tag{4}$$

setting $r_{cmin}$ and $r_{cmax}$ as the maximum and minimum radius of the grabbable area, respectively. For $l_{AB}sin\theta cos\beta \leq r_{cmin} \leq r_{cmax}$ or $r_{cmin} \leq r_{cmax} \leq l_{AB}sin\theta cos\beta$, the grasping error $E$ is:

$$E = \left| \sqrt{l_{AB}^2 + \left(\frac{r_{cmax}}{sin\theta}\right)^2 - 2l_{AB}\frac{r_{cmax}}{sin\theta}cos\beta - a^2} \right.$$
$$\left. - \sqrt{l_{AB}^2 + \left(\frac{r_{cmin}}{sin\theta}\right)^2 - 2l_{AB}\frac{r_{cmin}}{sin\theta}cos\beta - a^2} \right| \tag{5}$$

when $r_{cmin} \leq l_{AB}sin\theta cos\beta \leq r_{cmax}$, the grasping error $E$ is the maximum of $E_1$ or $E_2$.

$$E_1 = \left| \sqrt{l_{AB}^2 + \left(\frac{r_{cmax}}{sin\theta}\right)^2 - 2l_{AB}\frac{r_{cmax}}{sin\theta}cos\beta - a^2} - \sqrt{l_{AB}^2 sin^2\beta - a^2} \right| \tag{6}$$

$$E_2 = \left| \sqrt{l_{AB}^2 + \left(\frac{r_{cmin}}{sin\theta}\right)^2 - 2l_{AB}\frac{r_{cmin}}{sin\theta}cos\beta - a^2} - \sqrt{l_{AB}^2 sin^2\beta - a^2} \right| \tag{7}$$

If $r_{cmin}$ and $r_{cmax}$ are symmetrical to $r_c = l_{AB}sin\theta cos\beta$, then $E_1 = E_2$. The minimum grasping error could be obtained from Equations (6) and (7), thus:

$$l_{AB}^2 + \left(\frac{r_{cmax}}{sin\theta}\right)^2 - 2l_{AB}\frac{r_{cmax}}{sin\theta}cos\beta - a^2 = l_{AB}^2 + \left(\frac{r_{cmin}}{sin\theta}\right)^2 - 2l_{AB}\frac{r_{cmin}}{sin\theta}cos\beta - a^2 \tag{8}$$

$$\beta = cos^{-1}\frac{r_{cmax} + r_{cmin}}{2l_{AB}sin\theta} \tag{9}$$

To avoid structural interference of the two fingers during the reaching and clamping process:

$$L_v cos(\beta - \theta) < l_{AB} \tag{10}$$

For BC and BD to completely envelop the planting hole during the grasping process, the constraint conditions are:

$$2l_{AB}^2(1 - cos\gamma)sin\left(\frac{180° - \gamma}{2} - \beta + \theta\right) \geq r_g \tag{11}$$

$$a + l_{AB}sin(180° - \theta - \beta) \leq \frac{W_g}{2} \tag{12}$$

$$r_g \leq L_v sin\theta \leq \frac{L_g}{2} \tag{13}$$

### 2.2.3. Parameter Optimization of the Grabbing Mechanism

The main parameters affecting the grasping accuracy of the centering and grabbing mechanism are the finger length $l_{AB}$ and finger declination $\beta$ [26]. For the given $r_{cmin}$, $r_{cmax}$ of the grabbable area and dimension of the cultivation pipeline, finger length $l_{AB}$, and finger declination, $\beta$ should be optimized to meet the required grasping accuracy. Finger length $l_{AB}$, finger length increase step K, V-slot angle $2\theta$, and distance of the double-rotary pivot point 2a could be initially selected, and then the V-slot length $l_v$ could be determined with Equation (13), and iterative computation could be conducted with Equations (5)–(12). The calculation flow chart of n times of iterations is illustrated in Figure 9.

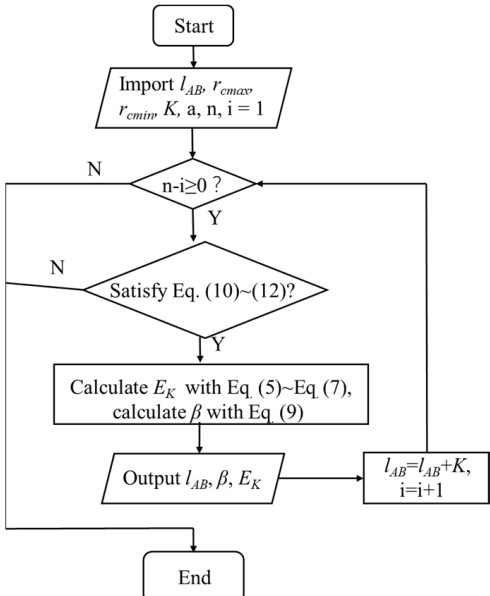

**Figure 9.** Flow chart of grabbing mechanism parameter optimization.

Less than 1mm of the grasping error is enough for harvesting the hydroponic Chinese kale [27]. On the basis of preliminary tests, $\theta$ is set as 60°, $a$ is 36mm, $K$ is 5mm, $n$ is 20, and initial $l_{AB}$ is 50mm. In Table 1, the $r_{cmin}$ and $r_{cmax}$ of the grabbable area of the hydroponic Chinese kale are 3.60 mm and 8.30 mm, respectively. The optimization program was run according to the flow chart in Figure 9, and iterative results of $l_{AB}$, $\beta$, and grasping error $E_k$ are shown in Table 2. It shows that with an increase in the iteration times, the grasping error decreases; after the seventh iteration, the grasping error reduces to less than 1mm.

According to structural dimension and grasping accuracy requirements, the optimal $l_{AB}$ and $\beta$ are 85 mm and 107.4°, respectively, and the grasping error is 0.96 mm.

**Table 2.** Calculation results of the finger parameters and grabbing error.

| Number of Iterations | Finger Length $l_{AB}$ (mm) | Finger Deflection $\beta$ (°) | Grasping Error $E_k$ (mm) |
|---|---|---|---|
| 1 | 55 | 117.6 | 1.69 |
| 2 | 60 | 115.1 | 1.49 |
| 3 | 65 | 113.0 | 1.34 |
| 4 | 70 | 111.3 | 1.22 |
| 5 | 75 | 109.8 | 1.12 |
| 6 | 80 | 108.5 | 1.03 |
| 7 | 85 | 107.4 | 0.96 |
| 8 | 90 | 106.4 | 0.90 |
| 9 | 95 | 105.5 | 0.84 |
| 10 | 100 | 104.7 | 0.79 |
| 11 | 105 | 104.0 | 0.75 |
| 12 | 110 | 103.4 | 0.71 |
| 13 | 115 | 102.8 | 0.68 |
| 14 | 120 | 102.2 | 0.65 |
| 15 | 125 | 101.7 | 0.62 |
| 16 | 130 | 101.3 | 0.59 |
| 17 | 135 | 100.9 | 0.57 |
| 18 | 140 | 100.5 | 0.55 |
| 19 | 145 | 100.1 | 0.53 |
| 20 | 150 | 99.8 | 0.51 |

Since the grabbable area height of the hydroponic Chinese kale shown in Table 1 is 25.03 mm, the height of the single middle layer is 10 mm, the height of the upper lower layer of the double-layer finger is 5 mm, and the height of the double-layer finger's hollow area is 15 mm. The total height of the grasping mechanism is 25 mm. The V-slot angle $2\theta$ was taken as 120°, and the largest finger rotation angle $\gamma$ in the clamping process was 30° according to Equation (1). As the maximum mass of the Chinese kale was 93.8 g, a 42-screw stepper motor with 40 mm of stroke and 600 g of the maximum load was selected to drive the two V-shaped fingers. It could achieve 0° to 77° of the finger rotation angle range, satisfying the required 30° of the finger rotation range. To avoid the two fingers' front surface directly impacting the leaves of the hydroponic Chinese kale, a conical arc surface structure was designed on the front end of the two fingers, as shown in Figure 10.

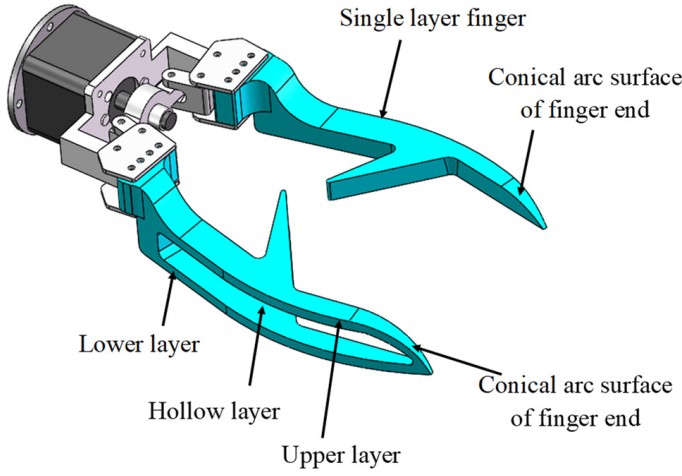

**Figure 10.** Structure diagram of the designed grabbing mechanism.

### 2.3. Grasping Test

#### 2.3.1. Test Platform

The grasping test platform consists of a conveyor belt, robotic arm, grabbing mechanism, fixed cultivation pipeline, high-speed camera, fill light, and laptop computer, as shown in Figure 11. The designed centering and grabbing mechanism is installed on a 3-axis 57 industrial robotic arm (Foshan Zhiqing Co., Ltd., Foshan, China). The motion ranges of the arm in the XY plane and along the Z axis are 0 to 1000 mm and −327.8 mm to 327.8 mm, respectively. Its motion speed is adjustable from 0 to 300°/s, repeat positioning accuracy is ±0.3 mm, total weight is 30 kg, and the maximum load is 2.5 kg. The arm is fixed on a conveyor belt, which could transport the grabbed hydroponic Chinese kale. Two sets of MV-CA016.10UC color industrial high-speed cameras (Hikvision Co., Ltd., Hangzhou, China) were selected to capture the grabbing process, which uses an IMX273 camera sensor with $1440 \times 1080$ mm of resolution and 166 fps of the maximum frame rate. The two high-speed cameras were used to capture the position and orientation of the kale in the $X_g Z_g$ and the $Y_g Z_g$ plane, respectively. A 30 W fill light was used to improve the video capture effect. The high-speed camera software MVSV3.1.0 was installed in the ASUS LAPTOP-LDHF8K7R laptop to record and process the captured video.

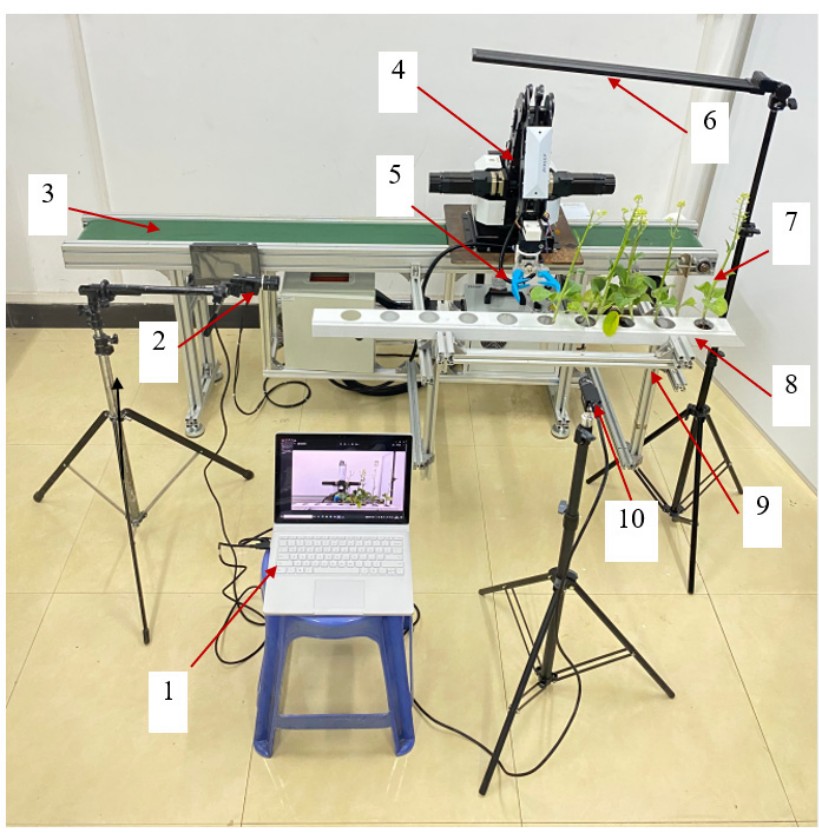

**Figure 11.** Grasping test platform. 1. Laptop; 2. no.1 high-speed camera; 3. conveyor belt; 4. robotic arm; 5. grabbing mechanism; 6. fill light; 7. hydroponic Chinese kale; 8. cultivation pipeline; 9. brackets for cultivation pipeline; 10. no.2 high-speed camera.

#### 2.3.2. Test Method

In each grabbing test, kale samples were placed in the planting holes with different initial positions and inclination angles. The robot arm was first programmed to open the two fingers of the grasping mechanism to align its BC edge (as shown in Figure 8) with the $X_g$ axis direction. Second, the two fingers were driven from the outside of the cultivation pipeline to the grasping position to make the clamping center of the two fingers consistent with the planting hole center. Fingers then closed at a certain deflection speed until the kale was clamped. Then, the kale was drawn out of the planting hole in the upward direction

along the $Z_g$ axis and rotated with the arm to the top of the conveyor belt. Finally, it was released on the convey belt and transported to the collector.

To investigate the effects of the enveloping contact and grasping action of the grabbing mechanism on the hydroponic Chinese kale with different initial inclination angles and at various positions of the planting hole, we selected the initial position of the kale relative to the planting hole $P_i$, its initial inclination angle $\Phi_o$, and the finger deflection speed $\omega$ as experimental factors. Eight $P_i$ (as shown in Figure 12a) was designed with 45° intervals, and four levels of the $\Phi_o$ (as shown in Figure 12b), 30~45°, 45~60°, 60~75°, and 75~90°, were defined. The five levels of $\omega$ were set as 20° s$^{-1}$, 40° s$^{-1}$, 60° s$^{-1}$, 80° s$^{-1}$, and 100° s$^{-1}$. The factor level table of the grabbing test is shown in Table 3.

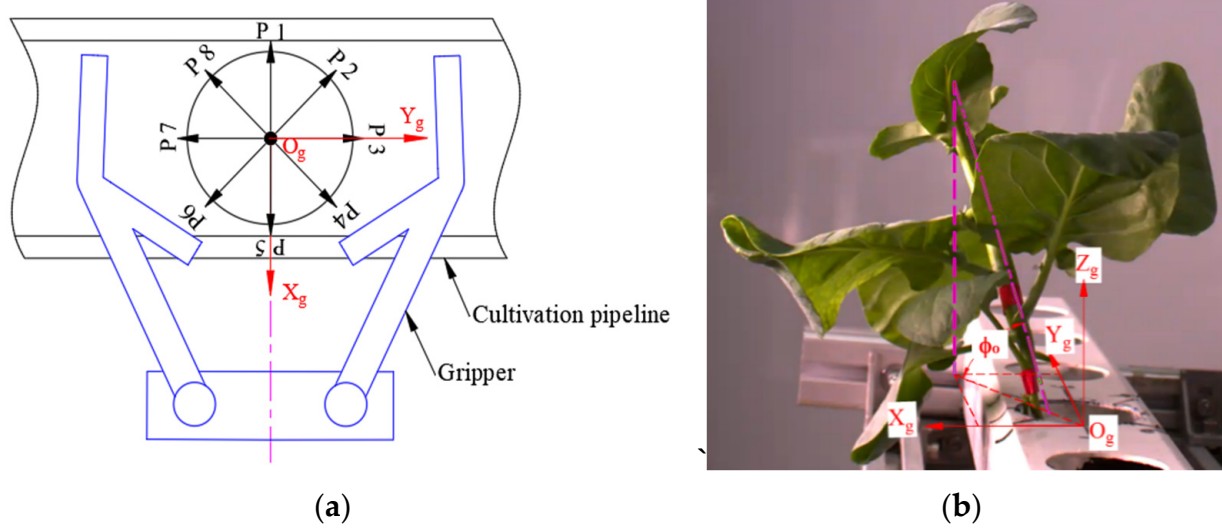

**(a)**            **(b)**

**Figure 12.** The initial posture of the test hydroponic Chinese kale: (**a**) the different initial position $P_i$, (**b**) the definition of the initial inclination angle $\Phi_o$.

**Table 3.** Factors and levels of grabbing test.

| Factor \ Level | Initial Position $P_i$ | Initial Inclination Angle $\Phi_o$ (°) | Finger Deflection Speed $\omega$ (° s$^{-1}$) |
|---|---|---|---|
| 1 | P1 | 30~45 | 20 |
| 2 | P2 | 45~60 | 40 |
| 3 | P3 | 60~75 | 60 |
| 4 | P4 | 75~90 | 80 |
| 5 | P5 | | 100 |
| 6 | P6 | | |
| 7 | P7 | | |
| 8 | P8 | | |

For conveniently observing the kale's position and orientation change in the grabbing process, red point A was marked on the bottom area of the kale, and red point B was marked around its mass center, as shown in Figure 13a. Videos of the grabbing process taken by the two high-speed cameras were imported into the motion analysis software Molysis. The image size was calibrated according to the 50 mm planting hole diameter. The marked red points A and B were set as tracking points. The two tracking points were searched along the $Z_g$ axis direction until the grabbing process was finished.

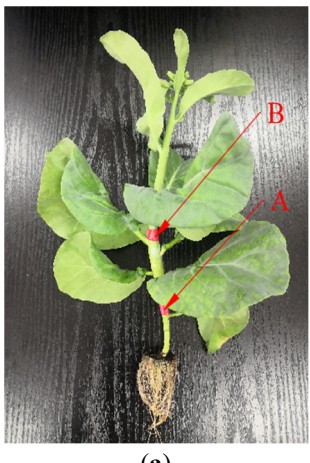
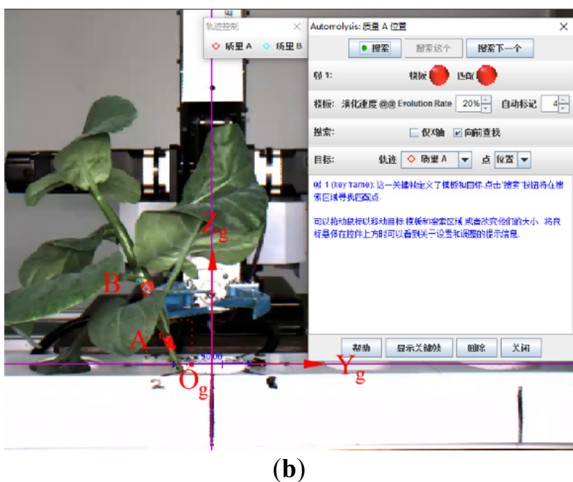
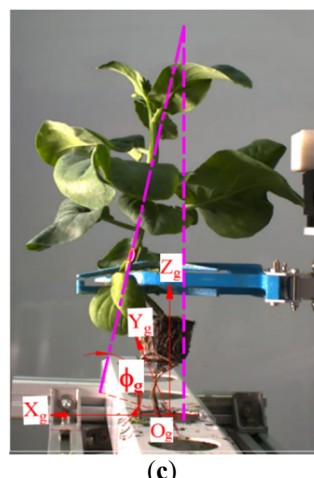

(**a**)          (**b**)          (**c**)

**Figure 13.** Data collection method in grasping test: (**a**) the illustration of red points A and B marked on the hydroponic Chinese kale, (**b**) the illustration of parameters setting interface in the motion analysis software Molysis, and (**c**) the definition method of the grasping inclination angle $\Phi_g$ in the grabbing test.

The three-dimensional spatial position coordinates of marked red point A and point B in the grabbing process were labeled $(x_A, y_A, z_A)$ and $(x_B, y_B, z_B)$, respectively, and the position coordinate data of point A and point B were the output. The parameter-setting interface in Molysis motion analysis software is shown in Figure 13b. After the hydroponic Chinese kale was grabbed out of the cultivation pipeline, the angle between the kale stalk and the $X_g Y_g$ plane was defined as the grasping inclination angle $\Phi_g$ of each test, as shown in Figure 13c. The closer $\Phi_g$ is to 90°, the better the centering and grasping effect is. On the basis of the output position coordinate data of the marked red point A and point B, $\Phi_g$ could be calculated with the following Equations (14) and (15). In Equation (15), $\vec{z}$ represents the unit vector of the $Z_g$ axis, and its position coordinate is (0, 0, 1):

$$\overrightarrow{AB} = \overrightarrow{O_gB} - \overrightarrow{O_gA} \tag{14}$$

$$\varnothing_g = 90° - \arccos\frac{\overrightarrow{AB} \cdot \vec{z}}{\left|\overrightarrow{AB}\right|\left|\vec{z}\right|} \tag{15}$$

A grabbing process without collision or interference from the grabbing mechanism, the hydroponic Chinese kale, or the cultivation pipeline is regarded as a successful grab. The successful and failed attempts are shown in Figures 14 and 15, respectively. For evaluating the grabbing ability of the designed mechanism, the grabbing success rate $P_s$ is defined by Equation (16):

$$P_s = \frac{n_s}{N} \times 100\% \tag{16}$$

where $n_s$ is the number of successful grabs, and $N$ is the total number of grabs. All factor tests were carried out three times. The average $\Phi_g$ and $P_s$ at different factor levels were compared by analysis of variance at the significance level of $p < 0.05$ (Duncan's method).

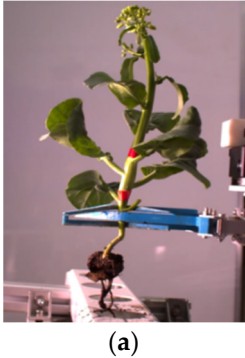
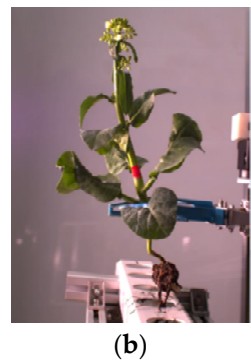
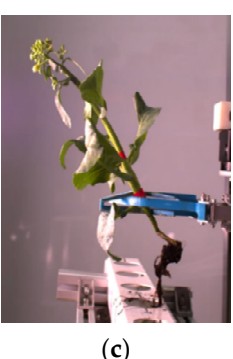

(**a**)          (**b**)          (**c**)

**Figure 14.** Successful grabbing attempts in the test: (**a**) the hydroponic Chinese kale was grabbed out of the cultivation pipeline in a fully vertical posture, (**b**) the hydroponic Chinese kale was grabbed out of the cultivation pipeline in a slightly inclined posture, and (**c**) the hydroponic Chinese kale was grabbed out of the cultivation pipeline in a relatively inclined posture.

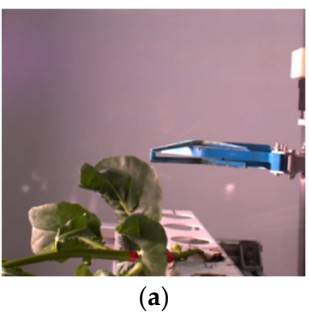
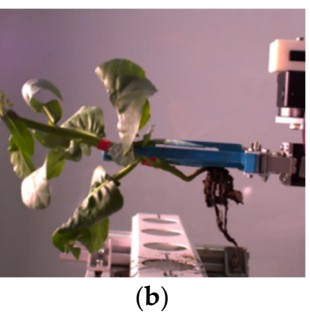
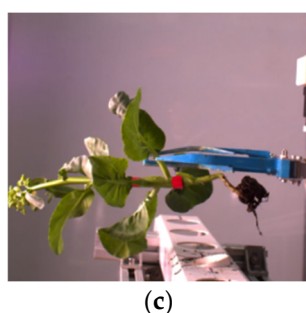

(**a**)          (**b**)          (**c**)

**Figure 15.** Failed attempts at grabbing kale in the test: (**a**) the hydroponic Chinese kale was not grasped, (**b**) the stalk of the kale was stuck in the finger, and (**c**) only the leaves of the hydroponic Chinese kale were grasped.

## 3. Results and Discussion

### 3.1. Results of Grasping Inclination Angle

The $\Phi_g$ results for 30° to 45° of $\Phi_o$ are shown in Figure 16a. At 20°s$^{-1}$ of $\omega$, 61.55° to 90.7° of $\Phi_g$ were obtained for different initial positions, and less than 80° of $\Phi_g$ were 62.5°, 65.9°, and 61.55° at the initial position of P$_1$, P$_2$, and P$_3$, respectively. At 40°s$^{-1}$ of $\omega$, $\Phi_g$ was in the range of 74.43° to 98.83° for different initial positions, and less than 80° of $\Phi_g$ was 74.43° at the initial position of P8. At 60°s$^{-1}$ of $\omega$, the obtained $\Phi_g$ were from 70.53° to 90.8° for different initial positions, and less than 80° of $\Phi_g$ was 70.53°, 79.8°, and 78.37° at the initial position of P1, P4, and P5, respectively. At 80°s$^{-1}$ of $\omega$, $\Phi_g$ was in the range of 61.8° to 96.85° for different initial positions, and less than 80° of $\Phi_g$ was 61.8°, 64.67°, 71.63°, 71.50°, and 77.40° at the initial position of P1, P2, P3, P4, and P8, respectively. At 100° s$^{-1}$ of $\omega$, $\Phi_g$ was in the range of 67.3° to 87.3° for different initial positions, and less than 80° of $\Phi_g$ was 66.97°, 73.50°, 78.07°, and 67.3° at the initial positions of P1, P6, P7, and P8. At the 40° s$^{-1}$ and 60° s$^{-1}$ of $\omega$, all obtained $\Phi_g$ was in the range of 70° to 100° for different initial positions. It indicated the centering grasping effects at 40° s$^{-1}$ and 60° s$^{-1}$ of $\omega$ are better than those at 20° s$^{-1}$, 80° s$^{-1}$, and 100° s$^{-1}$ of $\omega$.

The $\Phi_g$ results for 45° to 60° of $\Phi_o$ are shown in Figure 16b. At 20° s$^{-1}$ of $\omega$, the obtained $\Phi_g$ ranged from 74.23° to 88.07° for different initial positions, and less than 80° of $\Phi_g$ was 74.23° and 74.03° at the initial positions of P2 and P5, respectively. At 40° s$^{-1}$ of $\omega$, $\Phi_g$ was in the range of 70.7° to 93.13° for different initial positions, and less than 80° of $\Phi_g$ was 70.7° at the initial position of P1. At the 60° s$^{-1}$ of $\omega$, $\Phi_g$ was in the range of 73.85° to 89.1°, and less than 80° of $\Phi_g$ was 73.85° and 76.20° at the initial position of P5 and P7, respectively. At the 80° s$^{-1}$ of $\omega$, $\Phi_g$ was in the range of 60.55° to 88.03°, and less than 80° of $\Phi_g$ was 66.73°, 60.55°, and 74.93° at the initial position of P1, P2, and P5, respectively. At the 100° s$^{-1}$ of $\omega$, $\Phi_g$ was in the range of 67.07° to 90.03°, and less than 80° of $\Phi_g$ was

$67.10°$, $67.07°$, and $67.03°$ at the initial position of P1, P2, and P8, respectively. At the $20°$ $s^{-1}$, $40°$ $s^{-1}$, and $60°$ $s^{-1}$ of $\omega$, all obtained $\Phi_g$ were in the range of $70°$ to $100°$ for different initial positions. This indicated that the centering grasping effects at $20°$ $s^{-1}$, $40°$ $s^{-1}$, and $60°$ $s^{-1}$ of $\omega$ are relatively better than those at $80°$ $s^{-1}$ and $100°$ $s^{-1}$ of $\omega$.

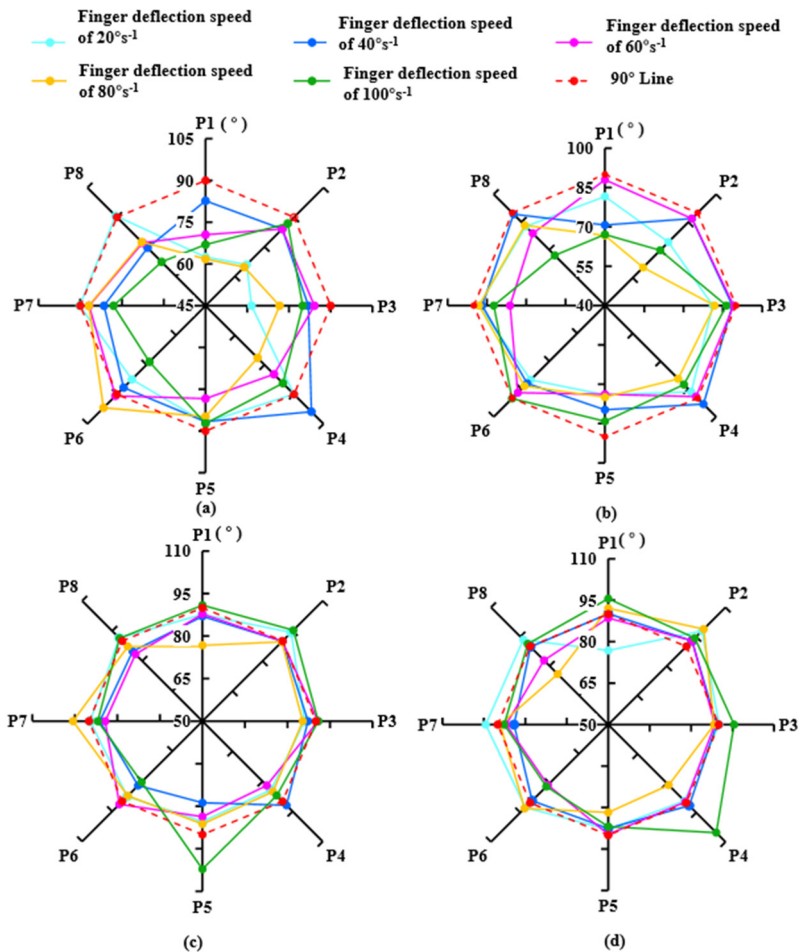

**Figure 16.** Test results for grasping inclination angle: (**a**) the results for $30°$ to $45°$ inclination angle at different initial position $P_i$ and different finger deflection speed, (**b**) the results for $45°$ to $60°$ inclination angle at different initial position $P_i$ and different finger deflection speed, (**c**) the results for $60°$ to $75°$ inclination angle at different initial position $P_i$ and different finger deflection speed, and (**d**) the results for $75°$ to $90°$ inclination angle at the different initial position of $P_i$ and different finger deflection speed.

The $\Phi_g$ results for $60°$ to $75°$ of $\Phi_o$ are shown in Figure 16c. At $20°$ $s^{-1}$ of $\omega$, all obtained $\Phi_g$ in the range of $84.23°$ to $94.27°$ were larger than $80°$ for different initial positions. At the $40°$ $s^{-1}$ of $\omega$, $\Phi_g$ was in the range of $78.77°$ to $91.93°$ for different initial positions, and less than $80°$ of $\Phi_g$ was $78.77°$ at the initial position of P5. At $60°$ $s^{-1}$ of $\omega$, all obtained $\Phi_g$ in the range of $82.07°$ to $91.43°$ were larger than $80°$ for different initial positions. At the $80°$ $s^{-1}$ of $\omega$, $\Phi_g$ was in the range of $76.73°$ to $95.67°$ for different initial positions, and less than $80°$ of $\Phi_g$ was $76.73°$ at the initial position of P1. At the $100°$ $s^{-1}$ of $\omega$, $\Phi_g$ was in the range of $80.37°$ to $102.7°$ for different initial positions, and $102.7°$ of $\Phi_g$ was obtained at the initial position of P8.

The $\Phi_g$ results for $75°$ to $90°$ of $\Phi_o$ are shown in Figure 16d. At the $20°$ $s^{-1}$ of $\omega$, $\Phi_g$ was in the range of $76.81°$ to $97.97°$ for different initial positions, and less than $80°$ of $\Phi_g$ was $76.81°$ at the initial position of P1. At $40°$ $s^{-1}$ and $60°$ $s^{-1}$ of $\omega$, $\Phi_g$ was in the range of $84.10°$ to $92.9°$ and $80.9°$ to $92.92°$ for different initial positions. At the $80°$ $s^{-1}$ of $\omega$, $\Phi_g$ was in the range of $75.87°$ to $98.87°$ for different initial positions, and less than $80°$ of $\Phi_g$ was

75.87° at the initial position of P1. At the 100° s$^{-1}$ of $\omega$, $\Phi_g$ was in the range of 81.70° to 105.3° for different initial positions, and more than 100 ° of $\Phi_g$ was 105.3° obtained at the initial position of P4. At the 40° s$^{-1}$ and 60° s$^{-1}$ of $\omega$, all obtained $\Phi_g$ were in the range of 80° to 100° for different initial positions. This indicated the centering grasping effects at 40° s$^{-1}$ and 60° s$^{-1}$ of $\omega$ were relatively better than those at 20° s$^{-1}$, 80° s$^{-1}$, and 100° s$^{-1}$ of $\omega$.

With the increase in $\Phi_o$, the $\Phi_g$ in different initial positions increased correspondingly, and the lowest $\Phi_g$ was larger than 60°. Variance analysis results illustrated that the influence of $\Phi_o$ on $\Phi_g$ was highly significant ($p = 0.0001$). When the $\Phi_o$ of the hydroponic Chinese kale was larger than 60°, all obtained $\Phi_g$ at different initial positions and finger deflection speeds were more than 75°. At 20° s$^{-1}$, 40° s$^{-1}$, and 60° s$^{-1}$ of $\omega$, $\Phi_g$ was mostly in the range of 85° to 95°. At the 80° s$^{-1}$ and 110° s$^{-1}$ of $\omega$, $\Phi_g$ was mostly in the range of 80° to 110°.

The statistical analysis indicated the influence of $\omega$ on $\Phi_g$ was insignificant ($p = 0.056$). The $\Phi_g$ in different initial positions at 40° s$^{-1}$ and 60° s$^{-1}$ are relatively better than those at 80° s$^{-1}$ and 100° s$^{-1}$ of $\omega$. It was observed that the contact impact of the two fingers on the initial static hydroponic Chinese kale at a low speed $\omega$ was smaller, and the bottom stalk of hydroponic Chinese kale could gradually move toward the center of the planting hole by the contact constraints. While at high speed of $\omega$, the contact impact of the two fingers on the initial static hydroponic Chinese kale was larger, and the change in the initial position of the hydroponic Chinese kale occurred suddenly, negatively affecting the centering and grasping action. To reduce the adverse influence of the initial contact, the start and stop deflection speed control strategies of the two fingers will be further studied.

These results revealed that the initial position of the hydroponic Chinese kale was a significant influence factor on the $\Phi_g$ ($p = 0.0208$). For different $\Phi_o$ and $\omega$, all $\Phi_g$ at the initial position of P1, P2, P5, and P8 were relatively lower. The reason might be that the initial BC edges of the two fingers are in the direction of the $X_g$ axis, and since P1 and P5 are in the front and back of the two fingers, the inner surfaces of the two fingers could not form centripetal grasping contact, securing the hydroponic Chinese kale at the initial position of P1 and P5. In addition, in the initial stage of grasping, centripetal contact constraints of the two fingers' inner surfaces on the Chinese kale at the initial positions of P2 and P8 were relatively poor. To achieve better and more secure contact with the kale at different initial positions, a new type of grasping mode that rotates the two fingers relative to the center of the planting hole to change the initial kale positions during the grasping process would be further explored.

### 3.2. Results of the Grabbing Success Rate

The grabbing success rate $P_s$ at different initial positions is shown in Figure 17. The $P_s$ at the initial positions P1, P2, and P3 was less than 90%. The lowest 80% of $P_s$ was obtained at the initial position P1, and $P_s$ of 85% was obtained at the initial position P2 and P3. The highest $P_s$ of 95% was obtained at the initial positions P4, P6, and P8, and $P_s$ of 90% was obtained at the initial positions P5 and P7. Variance analysis showed that the influence of the initial position $Pi$ on $P_s$ is significant ($p = 0.0225$). The grabbing test indicated that the initial positions P1, P2, and P3 were in the front envelope range of the double-layer finger. As the height of the double-layer finger is larger than that of the single-layer finger, the extended leaves of the hydroponic Chinese kale were more vulnerable to friction and collision from the front conical arc surface of the double-layer finger, which would cause the kale to incline to the front of the finger, resulting in grasping failure, as shown in Figure 15. Therefore, the front structure of the finger should be further optimized to reduce direct contact and collision with the extended leaves.

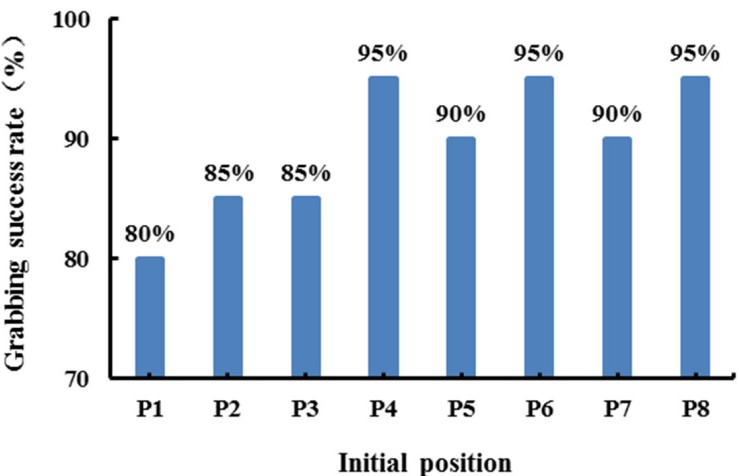

**Figure 17.** Grabbing success rate at different initial position $P_i$.

The grabbing success rate $P_s$ at different initial inclination angles, $\Phi_0$, are shown in Figure 18. Variance analysis results show that the $\Phi_0$ of the hydroponic Chinese kale is an extremely significant influence factor on the $P_s$ ($p = 0.0001$). With an increase in $\Phi_0$, $P_s$ increased correspondingly. At the $\Phi_0$ of 30~45°, 45~60°, 60~75°, and 75~90°, achieved $P_s$ was 72.5%, 85%, 97.5%, and 100%, respectively. It was found that the bottom leaves of the hydroponic Chinese kale with less than 60° of $\Phi_0$ are more likely to extend on the top surface of the cultivation pipeline and thus be much easier to push down by the entering fingers, which would lead to more grabbing failures. In the actual greenhouse planting conditions, the bottom extended leaves of mature hydroponic Chinese kale in adjacent planting holes offer mutual support, and the $\Phi_0$ of mature kale is mostly greater than 60°. To obtain more than 60° and consistent inclination angle of the mature hydroponic Chinese kale, the planting technology and management method should be further improved.

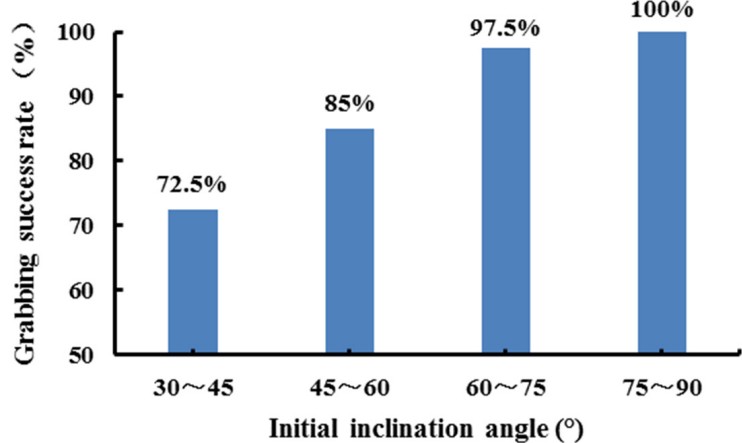

**Figure 18.** Grabbing success rate at different initial inclination angles.

The grabbing success rate $P_s$ at different finger deflection speeds, $\omega$, is shown in Figure 19. Less than 95% of $P_s$ were achieved at both the 20° s$^{-1}$ of $\omega$ and 80° s$^{-1}$ of $\omega$, and were 92.71% and 94.79%, respectively. The highest 98.96% of $P_s$ was at the 40° s$^{-1}$ of $\omega$. At 60° s$^{-1}$ and 100° s$^{-1}$ of $\omega$, the $P_s$ was 96.88% and 95.83%, respectively. Variance analysis results indicated that the influence of finger deflection speed on the grabbing success rate was insignificant ($p = 0.5427$). Suitable finger deflection speed for grabbing hydroponic Chinese kale should be in the range of 40° s$^{-1}$ to 60° s$^{-1}$.

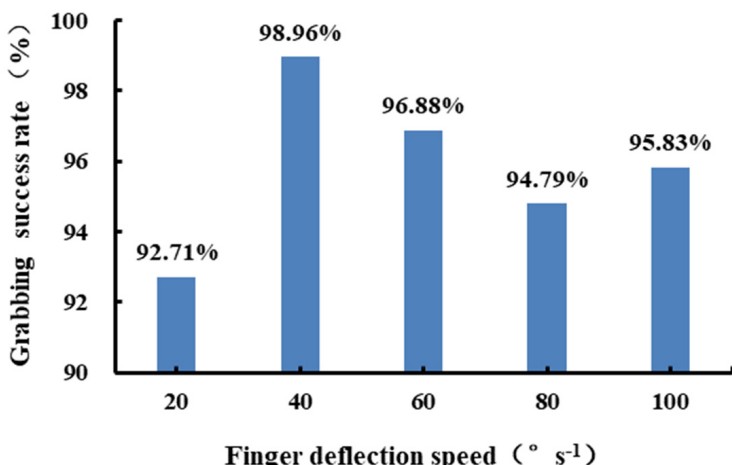

**Figure 19.** Grabbing success rate at different finger deflection speeds.

The operating time of each grabbing test was recorded by high-speed cameras. The results indicated that at $40° \ s^{-1}$ to $60° \ s^{-1}$ of finger deflection speed, the time for reach-in, grasping, and drawing out the hydroponic Chinese kale was about 3 s, which has no speed advantage compared with the manual method. As the center distance of adjacent planting holes on the cultivation pipe is the same, to improve the grabbing efficiency, a harvesting device was proposed with multiple grabbing mechanisms set at the end of the robot arm, which could synchronously grab multiple leaf vegetables at one time [28]. Its practical applicability will be validated in a future study.

### 4. Conclusions

On the basis of the planting environment and growth posture analysis and measurement of the grabbing-related feature parameters, an envelope centering and grabbing mechanism with double-pivot rotary V-shaped cross fingers for harvesting hydroponic leafy vegetables was developed in this study. Finger length and the deflection angle for grasping hydroponic Chinese kale were determined as 85 mm and 107.4°, respectively, on the basis of optimized objectives of less than 1mm of grasping error, sufficient envelope range, and small collision probability with the bottom leaves of the hydroponic Chinese kale.

A laboratory grabbing test was carried out by assembling the grabbing mechanism at the end of a three-axis 57 industrial robotic arm. The test results showed that the developed grabbing mechanism could obtain an 85° to 95° grasping inclination angle and more than 95% grabbing success rate at $40° \ s^{-1}$ to $60° \ s^{-1}$ of finger deflection speed for the matured hydroponic Chinese kale with more than 60° of initial inclination angle and different initial position in the planting hole. It proved the applicable centering and grabbing ability of the mechanism and indicated that the grabbing mechanism assembled on the robot arm has the potential to reduce labor intensity for harvesting the hydroponic Chinese kale.

To improve the feasibility of the grabbing mechanism in everyday greenhouse production conditions, new centering and grabbing motion patterns of the two fingers, the structural optimization of the finger front surface, suitable finger deflection speed control strategies, and Chinese kale planting technology will be studied in the future. The performance of the grabbing mechanism on other varieties of hydroponic leafy vegetables with similar growth characteristics should also be further verified.

**Author Contributions:** All authors contributed to the research. Conceptualization, H.X. and Y.L.; methodology, H.X.; validation, J.F., L.J., L.L., Z.D. and J.Z.; formal analysis, K.Z.; investigation, Z.D.; data curation, L.J..; writing—original draft preparation, Y.L.; writing—review and editing, H.X.; visualization, J.Z.; supervision, J.F.; project administration, K.Z.; funding acquisition, H.X. All authors have read and agreed to the published version of the manuscript.

**Funding:** This research was funded by the "Guangdong Natural Science Foundation of China (Grant No.: 2021A1515010777) and the "Guangdong Modern Agricultural Industrial Technology System Innovation Team Project of China (Grant No.: 2019KJ131)".

**Data Availability Statement:** Data are contained within the article.

**Acknowledgments:** The authors thank the experts for editing our paper and the anonymous reviewers for their critical comments and suggestions to improve this article.

**Conflicts of Interest:** The authors declare no conflict of interest.

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
