# Peer review of "An Enveloping, Centering, and Grabbing Mechanism for Harvesting Hydroponic Leafy Vegetables Cultivated in Pipeline"

_agronomy, doi:10.3390/agronomy13020476_

Round 1

Reviewer 1 Report

In this paper, a novel grabbing mechanism with double pivot rotation cross fingers was proposed, which could grasp hydroponic leafy vegetables in different positions and planting hole directions. The grabbing success rate was high, which was innovative in the field of hydroponic leafy vegetables harvesting. The following parts of this article need improvement:

1. Why are the contact forces of single finger and double finger divided into upper and lower layers? (Lines 185-186).

2.FP 1 is inconsistent with the description in the paper (Figure. 6b). It is suggested to mark the torque center in the figure (FIG. 6).

3. The meaning of each character in the text is not fully reflected in the figure (Figure 8) (Formula 1-13).

4. Uneven distribution of stems and leaves at the bottom of vegetables. Will the clamping process of the mechanism be disturbed by this phenomenon?

5. If the stem diameter of the working object is different, will the small closing Angle lead to the deformation of the valve stem?

Reviewer 2 Report

In the paper, a grabbing mechanism with double pivot rotary V-shaped cross fingers was proposed, and its main parameters was optimized for harvesting hydroponic leaf vegetable. Through laboratory test, it showed promising applicability for harvesting hydroponic Chinese kale. While some grammar problems should be checked in the manuscript, and the conclusion should be revised.

(1) Line 33-34: Not clear, please list some references.

(2) Line 134: a digital electronic vernier calipers (range:500g,accuracy:0.01mm), which there maybe wrong for the unit expression.

(3) Line 165: fingers could apply translating contact constraints, please express more clearly.

(4) Many contents of the conclusion part are repetitions of the results part.

Reviewer 3 Report

This manuscript is “An envelope centering grabbing mechanism for harvesting hydroponic leaf vegetable cultivated in pipeline” (agronomy-2150713). Comments are as follows.

(1) How does the automatic grabbing device ensure no harm to aquatic plants?

(2) How does the author consider that plants will collapse during growth?

(3) This manuscript lacks the economic analysis and future industrialization prospect of the new equipment.

(4) Research significance needs to be added in the Introduction section.

(5) The manuscript lacks discussion chapters and needs to be supplemented.

(6) I suggest that the authors invite native speakers to revise the English.

(7) The reference in the main document of the manuscript is marked incorrectly. The format of references does not meet the requirements of the journal.

Reviewer 4 Report

Dear Authors,

The manuscript is interesting and presents a detailed evaluation of grabbing mechanism with theory. I have a few minor suggestions to improve your work.

1) The figure captions should be written in more detail. For example, in Fig 13, it is difficult to understand (a-c) figures. Similarly in Fig 16. For each figure, please elaborate the topic of the figure and what each sub-figure represents.

2) The Table captions also should be in detail. For example, Table 3 should state the topic of the Table in more detail.

3) For the grabbing mechanism in Fig 10, I am curious how only one motor can provide movements in x, y, z directions. Wouldn't you need two motors - one for opening of gripper and other for inclination?

4) Is the flowchart in Fig 9 used during the experimentation? How are the constraints satisfied during actual test?

5) There is a reference paper I can suggest that built a grasping gripper for sieves using motors. The gripper was integrated with the entire instrument. Please see this paper for ideas on how to show the gripping action in figure, and possibly add videos of your working gripper in action:

 Legner, C.M., Tylka, G.L. & Pandey, S. Robotic agricultural instrument for automated extraction of nematode cysts and eggs from soil to improve integrated pest management. Scientific Reports 11, 3212 (2021). https://doi.org/10.1038/s41598-021-82261-w

Reviewer 5 Report

1. Add contribution section after introduction part.

2. Detailed discription required for all figures present in manuscript.

3.Provide dataset link

4.Comparison table required for both proposed and existing work.

5.Provide high resolution images, so images are look blur, example figure 2.

6. Add future work.

7.Refer recent reference to strengthen the related work

1.Wang, D., Su, R., Xiong, Y., Wang, Y., & Wang, W. (2022). Sugarcane-Seed-Cutting System Based on Machine Vision in Pre-Seed Mode. Sensors, 22(21), 8430.

2.Jayagopal, P., Rajendran, S., Mathivanan, S. K., Sathish Kumar, S. D., Raja, K. T., & Paneerselvam, S. (2022). Identifying region specific seasonal crop for leaf borne diseases by utilizing deep learning techniques. Acta Geophysica, 1-14.
